# Therapeutic Drug Monitoring of Tacrolimus-Personalized Therapy in Heart Transplantation: New Strategies and Preliminary Results in Endomyocardial Biopsies

**DOI:** 10.3390/pharmaceutics14061247

**Published:** 2022-06-12

**Authors:** Simona De Gregori, Annalisa De Silvestri, Barbara Cattadori, Andrea Rapagnani, Riccardo Albertini, Elisa Novello, Monica Concardi, Eloisa Arbustini, Carlo Pellegrini

**Affiliations:** 1U.O.C Laboratorio Analisi Chimico Cliniche, Fondazione IRCCS Policlinico San Matteo, 27100 Pavia, Italy; r.albertini@smatteo.pv.it (R.A.); elisa.novello1996@gmail.com (E.N.); 2U.O.S Epidemiologia Clinica e Biostatistica, Fondazione IRCCS Policlinico San Matteo, 27100 Pavia, Italy; a.desilvestri@smatteo.pv.it; 3U.O.C. di Cardiochirurgia, Fondazione IRCCS Policlinico San Matteo, 27100 Pavia, Italy; b.cattadori@smatteo.pv.it; 4Unità di Chirurgia Cardiaca, Dipartimento di Scienze Clinico Chirurgiche, Diagnostiche e Pediatriche, Università degli Studi di Pavia, 27100 Pavia, Italy; andrea.rapagnani01@universitadipavia.it; 5Centro Malattie Genetiche Cardiovascolari, Fondazione IRCCS Policlinico San Matteo, 27100 Pavia, Italy; m.concardi@smatteo.pv.it (M.C.); e.arbustini@smatteo.pv.it (E.A.); 6Unità di Chirurgia Cardiaca, Dipartimento di Scienze Clinico Chirurgiche, Diagnostiche e Pediatriche, Università degli Studi di Pavia—U.O.C. di Cardiochirurgia, Fondazione IRCCS Policlinico San Matteo, 27100 Pavia, Italy; carlo.pellegrini@unipv.it

**Keywords:** heart transplantation, acute rejection, therapeutic drug monitoring, tacrolimus, endomyocardial biopsies

## Abstract

Tacrolimus (TAC) is an immunosuppressant drug approved both in the US and in the EU, widely used for the prophylaxis of organ rejection after transplantation. This is a critical dose drug: low levels in whole blood can lead to low exposure and a high risk of acute rejection, whereas overexposure puts patients at risk for toxicity and infection. Both situations can occur at whole-blood concentrations considered to be within the narrow TAC therapeutic range. We assumed a poor correlation between TAC trough concentrations in whole blood and the incidence of acute rejection; therefore, we propose to study TAC concentrations in endomyocardial biopsies (EMBs). We analyzed 70 EMBs from 18 transplant recipients at five scheduled follow-up visits during the first year post-transplant when closer TAC monitoring is mandatory. We observed five episodes of acute rejection (grade 2R) in three patients (2 episodes at 0.5 months, 2 at 3 months, and 1 at 12 months), when TAC concentrations in EMBs were low (63; 62; 59; 31; 44 pg/mg, respectively), whereas concentrations in whole blood were correct. Our results are preliminary and further studies are needed to confirm the importance of this new strategy to prevent acute rejection episodes.

## 1. Introduction

Heart transplant (HTx) remains the gold standard treatment for end-stage heart failure. Survival has significantly increased up to 12 years [1] and improvements in immunosuppressive therapies are one of the factors that has contributed considerably to better outcomes. Tacrolimus (TAC), the primary immunosuppressive drug for HTx recipients, has proven to be superior to cyclosporine (CyA) in both the prevention and treatment of rejection [2]. The two drugs differ in their chemical structure (CyA is a cyclic endecapeptide, whereas TAC is a macrocyclic lactone) but they act in a similar way. They are calcineurin inhibitors and although their main mechanism of action is similar, TAC produces therapeutic effects at concentrations 100 times lower than CyA and, consequently, has a reduced risk of toxicity.

Tacrolimus administered orally is rapidly absorbed with a mean time to maximal concentration (t_MAX_) of 1–2 h, but the composition of food may highly influence its absorption. The highly lipophilic character of TAC largely explains this phenomenon. Another factor regulating TAC bioavailability is P-glycoprotein (Pgp), an efflux pump that is situated in the apical membrane of mature epithelial cells, in hepatocytes, in renal tubular cells, in the leucocytes, and also in the blood–brain barrier. When TAC passes Pgp and enters the enterocyte, it is metabolized by the cytochrome P-450 CYP3A. The expression of Pgp is influenced by genetics. The bioavailability of TAC has been found to be approximately 15% but may vary between 4 and 89%. TAC is highly bound to erythrocytes. It’s binding to plasma proteins varies between 72 and 98% depending on the methodology used. Because of the extensive partitioning of tacrolimus into erythrocytes, its apparent volume of distribution (Vd) based on blood concentrations is much lower (1.0 to 1.5 L/kg) compared with values based on plasma concentrations (about 30 L/kg). Patients treated with a calcineurin inhibitor (CNI) are at high risk of developing kidney injury; nephrotoxicity is manifested either as acute kidney injury (AKI), which is largely reversible after reducing the dose, or as chronic progressive kidney disease, which is usually irreversible. Other kidney effects of CNI include tubular dysfunction and, rarely, thrombotic microangiopathy (TMA) that can lead to acute kidney allograft loss after kidney transplantation.

Tacrolimus is a critical dose drug; low levels in whole blood can lead to low exposure and a high risk of acute rejection, whereas overexposure puts patients at risk for both toxicity (as reported above) and infections. Both situations can occur at whole-blood concentrations considered to be within the narrow TAC therapeutic range. Moreover, data from three randomized controlled trials did not find an association between TAC pre-dose concentrations (trough: C_0_), the reference parameter for therapeutic drug monitoring, and the incidence of acute rejection [3,4].

Tacrolimus demonstrates large pharmacokinetic inter-individual variability, partially due to pre-systemic metabolism by the intestinal cytochrome P450 (CYP3A4/5) [5], which may be affected by other CYP3A substrates, inducers, inhibitors, demographic characteristics, hepatic dysfunction, and hematocrit. High variability in TAC trough levels has been related to worse outcomes both in kidney transplant recipients [6] and in HTx populations [7,8,9], mostly during the first year after transplantation when closer TAC monitoring is mandatory (range: 5–20 ng/mL).

Rejection continues to be one of the leading causes of death during the first year after HTx [10]; the role and usefulness of endomyocardial biopsy (EMB) in routine surveillance during this period remain controversial. EMB should be performed to detect any evidence of graft rejection when non-invasive tests, such as cardiac magnetic resonance imaging (MRI) or positron emission tomography (PET) scans, cannot provide a diagnosis. Nevertheless, the need for frequent monitoring makes cumbersome the use of non-invasive tests.

We assumed a poor correlation between TAC trough concentrations in whole blood and incidence of acute rejection; therefore, we propose to study TAC concentrations in EMBs. To the best of our knowledge, we believe this is the first work showing preliminary results of TAC-concentration profiles (pg/mg vs. time) in EMBs of HTx patients at six scheduled follow-up visits during the first post-transplant year.

We want to share our preliminary results, although our study is ongoing and further analysis is required before drawing conclusions.

## 2. Materials and Methods

### 2.1. Patients

The aim of our study “Therapeutic Drug Monitoring of Tacrolimus Personalized Therapy in Heart Transplantation: new strategies” (cod. 08073421), approved by the ethics committee of “Fondazione IRCCS Policlinico San Matteo” and supported by 5 × 1000 donations, is to evaluate possible correlations between TAC concentrations in EMBs and acute rejection episodes in order to find an accurate, specific, and predictive marker.

The entire study, which started in 2020 and is ongoing,”, requires the enrollment of at least 25 de novo transplant recipients, male and female, aged 18 to 70, who receive TAC twice-daily dosing (BID) in combination with steroids and antiproliferative drugs (Mycophenolate Mofetil, Sodium Mycophenolate). To date, 33 patients (9F/24M) have been screened; 18 of them (6F/12M; median age 57 years old, max 69–min 23) underwent HTx and were enrolled in the study, whereas the other 15 are still on the waiting list. Within the group of patients who received the graft, 10 completed the follow-up period, 2 completed the sixth month, 4 completed the third month, and 2 died between the first and the third months. Each patient was anonymized and identified by the alphanumeric code TAC-XX, where XX is a consecutive number assigned during the enrollment.

All participants signed informed-consent forms, authorizing the use of their samples for the study. At each of the study time points (15 days, 1 month, 3 months, 6 months, and 12 months after transplantation), two whole blood samples (5 mL each) were collected in EDTA-containing tubes for TAC quantification both in whole blood and in the peripheral blood monocytes cells (PBMC—data not reported). EMBs from the transplanted heart were also obtained both for histopathological analysis and for TAC quantification (one EMB/each study time-point).

### 2.2. Sample Preparation

Tacrolimus whole-blood concentration was measured by antibody-conjugated magnetic immunoassays (ACMIAs) (Dimension instrument) from Siemens Healthcare Diagnostics, according to the manufacturer’s specifications and the clinical practice guidelines of the laboratory. Tacrolimus quantitation in cardiac biopsies was analyzed by a combined enzymatic-digestion/mass spectrometry assay based on an analytical method that we validated and published in September 2020 [11]. Briefly, the obtained EMBs were properly weighted and then incubated in 50 µL of digestion buffer (10% Proteinase K solution in TAC-free ATL buffer) at 55 °C for 90 min. In the end, the tissue was completely solubilized and was added with 20 µL of internal standard (FK-506-^13^CD_2_: 100 ng/mL), 300 µL of water, and 1 mL of tert-butyl methyl ether into the same cryovial for reaction. To promote the TAC extraction into the organic phase, samples were gently mixed for 15 min on a rotary mixer at room temperature to avoid an emulsion between the aqueous and organic layers and then centrifuged at 10,400× *g* for 10 min. In the end, the organic phase was evaporated to dryness under a gentle flow of nitrogen (room temperature). Dry residues were reconstituted by adding 60 µL of methanol and injected into the liquid chromatography–mass spectrometry (LC–MS/MS) system.

### 2.3. HPLC-MS/MS Assay

After the cleanup procedure, the extracted samples underwent an online solid phase extraction (SPE) coupled to liquid chromatography–tandem mass spectrometry; the required trap column and the analytical column (heated and maintained at 50 °C) were purchased from Chromsystems (REF 93110 and 93100, respectively). Elution was carried out in gradient mode at a flow rate of 0.5 mL/min with ammonium acetate 2 mM (acidified with 0.1% HCOOH) in water (mobile phase A) and ammonium acetate 2 mM (acidified with 0.1% HCOOH) in CH_3_OH (mobile phase B).

Ammonium acetate and formic acid promote the formation of the ammoniated precursor ions ([M + NH_4_]^+^, TAC *m*/*z* 821.3; ^13^CD_2_-TAC *m*/*z* 824.3) that can be easily fragmented. Multiple-reaction monitoring (MRM) mode was used to simultaneously detect TAC and its isotopic analog (^13^CD_2_-TAC), chosen as the internal standard; the optimal instrument parameters and MS/MS transitions (*m*/*z* 821.3→768.0; *m*/*z* 824.3→771.0) were determined by direct infusion at a flow rate of 7 µL/min for TAC and ^13^CD_2_-TAC separately into the mass spectrometer at a concentration of 1 µg/mL in mobile phase B.

### 2.4. Calibration Standards and Quality Controls (Qc)

Calibration curves were prepared by spiking the target analyte (TAC) into a matrix that has been judged to be representative of the real samples’ matrix. We bought a swine heart in a butcher’s shop and cut it into many small pieces (0.5–5 mg); each section was added with 50 µL of digestion buffer standard solution containing TAC at known concentrations (standard solutions). Five calibrators and 3 quality controls have been prepared and used to obtain the daily calibration curve; the peak area ratios of TAC to IS (Area_TAC_/Area_IS_) were plotted as a function of the quantity of TAC (ng) added to the pieces of swine heart (TAC [ng] = standard solution concentration [ng/mL] × 50 × 10^−3^ mL).

We prepared calibration standards and quality controls containing 0.033, 0.065, 0.130, 0.260, 0.520 ng and 0.070, 0.208, 0.416 ng of TAC, respectively. The deviation of standards from their nominal values could not exceed 15% (20% for the Lower Limit of Quantification: LLOQ, 0.033 ng).

### 2.5. Data Analysis

The Xcalibur 2.07 and LCquan 2.5.6 software (Thermo Fisher Scientific, San Francisco, CA, USA) were used for LC–MS/MS system control, data acquisition, and data analysis; the calibration curves were established by plotting the peak area ratio (analyte/IS) versus the TAC nominal concentrations or nominal added quantity in blood or in EMBs respectively, using a weighted (1/x) linear regression curve. Analyte peaks were identified with a combination of retention times and the specific MRM transition.

The amount of TAC (ng) in patients’ biopsies was back-calculated from the daily calibration curve equation; the TAC concentration in EMBs was expressed as the ratio between the measured TAC amount and the initial EMB weight (mg) (Equation (1)).
(1)$${\left[\mathrm{TAC}\right]}_{\mathrm{EMB}}\;=\;\frac{\mathrm{Amount}\;\mathrm{of}\;\mathrm{TAC}\;\left(\mathrm{ng}\right)}{\mathrm{weight}\;\mathrm{of}\;\mathrm{EMB}\;(\mathrm{mg})}\;\times \;1000\;=\;\left[\frac{\mathrm{pg}}{\mathrm{mg}}\right]$$

### 2.6. Rejection Surveillance

#### 2.6.1. Technical Considerations

Protocol-based EMBs are performed in the first year after HTx for acute rejection surveillance, according to our institute’s good clinical practice.

Each EMB is assessed for acute cellular rejection (ACR) and antibody-mediated rejection (AMR) as stated by the International Society for Heart and Lung Transplantation (ISHLT) guidelines. The presence of circulating donor-specific antihuman leukocyte antigen (HLA) antibodies (DSAs) is considered a mandatory criterion for AMR after HTx. DSAs are known as prognostic biomarkers of outcome; recipients with de novo DSA have a threefold increased risk of mortality [12].

Endomyocardial biopsy (EMB) is a valuable diagnostic tool for myocardial disease, for monitoring cardiac allograft rejection, and for diagnosing inflammatory and infiltrative cardiomyopathies. At our center, an experienced cardiothoracic surgeon uses a disposable rigid bioptome inserted through the right internal jugular vein and guided into the right atrium to cross the tricuspid valve and reach the right ventricular septum under echocardiographic guidance. The benefit of this new biopsy catheter was adequate endomyocardial sampling without procedure-related complications [13].

The interventricular septum is the preferred biopsy site for its thickness, compared to the free wall of the right ventricle for its continuity with the left ventricle and for its location in the natural path of the blood flow, which facilitates access. The drawback is that repeated EMB sampling results in a restricted region of the endocardium being assessed and may result in interpretive errors.

The tissue sample procured is usually a 1–2 mm cube of endocardium and myocardium. After extraction of the tissue fragment, the cardiovascular surgeon was careful not to remove the specimen with forceps, but rather to gently “move it” with a needle from the biopsy catheter and place it directly into 10% neutral-buffered formalin. The fixative was kept at room temperature to prevent additional contraction band artifacts. Histological preparation, embedding, staining, reading, and reporting of the diagnosis within 24 h are standard procedures.

The cardiovascular surgeon used the same careful approach for the biopsies intended for the determination of the TAC concentration; he placed them in empty cryovials and immediately called the laboratory for the subsequent standardized procedures (weighing and storage at −80 °C until analysis).

#### 2.6.2. Histological Preparation

The microscopic description of acute cellular rejection in the cardiac allograft is generally accepted as the presence of a myocardial mononuclear infiltrate, hemorrhage, myocyte injury or necrosis, and vascular endothelial lesions, which includes endothelial disruption and platelet fibrin deposition. The sensitivity of detecting transplant rejection can approach 98% with five adequate biopsy fragments, yet more than six samples do not appear to increase diagnostic yield [14]. The greatest potential limitation to EMB interpretation is sampling error. The adequacy of tissue fragments is very important for correct diagnostic accuracy and interpretation.

Standard histological preparation requires paraffin-wax embedding followed by ribbons of 4 µm thick sections mounted on glass slides. Slides are numbered sequentially and stained with hematoxylin and eosin for histomorphological characterization.

Evaluation of sample adequacy for the International Society of Heart and Lung Transplantation grading scheme requires a minimum of four good endomyocardial tissue fragments, with less than 50% of each fragment being fibrous tissue, thrombus, or other non-interpretable tissue fragments (such as crush artifacts or poorly processed fragments).

### 2.7. Statistical Analysis

Medians are presented in graphs showing a single outcome measured at several points over time. Points are connected by straight lines. Error bars show an interquartile range (IQR). Joint modeling would be more appropriate considering the association between time-to-event (acute rejection episode) and the measured longitudinal data (TAC concentrations). However, in our opinion, the limited sample size (5 events, 3 patients) is not adequate for estimating the effects of the longitudinal process in joint modeling [15].

## 3. Results

### 3.1. Concentration–Time Profiles

Figure 1, Figure 2, Figure 3 and Figure 4 show the concentration–time profiles and the median concentration time-profiles of TAC in EMBs and whole blood samples, respectively, from each of the 18 transplant recipients.

The analysis in the two different matrices was always performed on the same day to reduce possible and unexpected interferences.

TAC concentrations in EMBs were variable, especially during the first months after transplantation; at the first (15 days) and second follow-up visits (1 month), TAC_EMB_ of patients TAC-04 and TAC-21 were unexpectedly elevated (207 and 257 pg/mg, respectively) but over the next few months their values dropped to conform to the others.

The maximum concentration value was detected around 1 month after transplantation (257 pg/mg; patient TAC-21), whereas the minimum level (18 pg/mg) was observed 15 days after HTx (18 pg/mg; patient TAC-11), as reported in Table 1.

At the third evaluation (3 months), some TAC_EMB_ profiles reached a minimum and then increased again in the following months. The results corresponding to 1 year post-transplantation come from a low number of samples (Table 1: N = 9) but a decreasing trend is plausible. At this time point, the dispersion of data appears minimal but only half of the patients completed the observation period. All these results are better summarized in Figure 2, where the median concentration–time profile of TAC_EMB_ and the related interquartile range (IQR) are reported as time functions (months). Although the pattern is very irregular, the IQR is overlapping.

In contrast, the concentration–time profiles of TAC_WB_ during the entire first year post-HTx were more regular for all patients (Figure 3), as required by the universally accepted therapeutic drug monitoring guidelines.

The corresponding calculated median concentration–time profile shows a continuous slight increase (8.2–13.8 ng/mL), although the interquartile ranges (IQR) are overlapping (Figure 4). Fluctuations are limited and always within the expected therapeutic range (5–20 ng/mL).

When the study started, the potential concentration range for TAC in EMBs was unknown; Capron and colleagues [16] found TAC levels ranging from less than 5 up to 387 pg/mg in liver biopsies. The tissue levels displayed an excellent correlation with the liver histopathologic BANFF rejection score, whereas the blood levels did not.

Even though the liver and heart are obviously different organs, Capron’s work was a good starting point for the present study; to date, similar concentrations were found in EMBs ranging from 18 to 257 pg/mg.

### 3.2. Acute Rejection Episodes

Five acute rejection episodes (grade 2R) [17] were observed in three patients, whose characteristics are summarized in Table 2; 1 episode occurred at 0.5 months, 3 at 3 months, and 1 at 12 months, when the TAC concentrations in EMBs were low (63; 62; 59; 31; 44 pg/mg, respectively).

The highest ratio of TAC_WB_/TAC_EMBS_ was reached 3 months after HTx, when TAC_WB_ was always in the expected therapeutic range, whereas the TAC_WB_/TAC_EMBs_ ratio was the lowest in the presence of acute rejection (Table 3).

Despite the small sample size, TAC concentrations were analyzed in both EMBs and whole blood by considering the results of rejecting (RPs) and non-rejecting patients (NRPs) separately; for the second time, the conclusions are different depending on the specific matrix.

The median TAC concentrations in EMBs (Figure 5) and whole blood (Figure 6) of RPs and NRPs were plotted; in both situations the IQRs overlap, but in Figure 5 the median concentration of the RP group is lower, whereas in Figure 6 the opposite situation occurs.

## 4. Discussion and Conclusions

Despite the progress and improved overall outcomes, acute allograft rejection (AR) remains the Achilles’ heel of heart transplantation. The manifestations of rejection can occur as early as intraoperatively to many years after transplant. The timing of AR plays a significant role in establishing cause and diagnosis. Acute rejection can either be responsible for early graft dysfunction, occurring in the first days after surgery, or late graft dysfunction developing weeks to years after transplantation.

Acute allograft rejection is an important contributor to graft failure, which remains a leading cause of death (10%) within the first three years after HTx; low TAC levels in whole blood can lead to low exposure and an increased risk of acute rejection, but, unfortunately, an acceptable correlation between these two factors has never been demonstrated.

Episodes of acute rejection can also occur when TAC concentrations fall within their narrow therapeutic range. The unbound concentration has been shown to be a crucial factor in cellular uptake and may increase glomerular vasoconstriction leading to nephrotoxicity in the early days after transplantation. From a mechanistic point of view, the plasma concentration of unbound TAC is a more reasonable parameter to monitor to achieve optimal TAC dosing in transplant patients, especially in the early days after HTx, but current assays used for routine TAC monitoring lack the sensitivity to adequately measure it [2].

From 1995 to 2021, many clinical studies [18] investigated whole blood, PBMC, and allograft TAC concentrations and their association with clinical outcomes, to evaluate an evident clinical benefit with respect to the prediction of rejection. All the studies were conducted on liver- or kidney-transplanted patients; none involved HTx patients. The results are controversial; well-designed and powered prospective clinical trials are still needed to determine whether TAC therapeutic drug monitoring (TDM) in alternative matrices offers a significant clinical benefit over the current TDM based on whole blood determinations.

In order to collect initial data from HTx patients, preliminary results on TAC concentration-time profiles both in whole blood and in EMBs during the first year after transplantation are presented. Each specialized center normally determines the minimum levels of the appropriate immunosuppressant drug in whole blood to prevent AR episodes; in general, target TAC concentrations are highest soon after HTx and slowly decrease over the first year, eventually settling on the lowest maintenance levels of immune suppression that are compatible with AR prevention and the attenuation of drug toxicities. Another general principle is to favor the use of low doses of multiple drugs without overlapping toxicities over the use of higher doses of fewer drugs whenever feasible. A third principle is that excessive immunosuppression is undesirable because it leads to a high incidence of side effects such as infections and malignancies. Finding the right balance between over- and under-immunosuppression in an individual patient is truly an art that requires science. The reason for the therapeutic choice by our center was poor post-operative peripheral perfusion, which requires a lower dosage of nephrotoxic drugs to not overload the kidneys. As summarized in Table 3, TAC_WB_ concentrations were maintained as constant throughout the observation period and, although the median was significantly lower than the others 15 days after HTx, it was still within the expected therapeutic range (5–20 ng/mL).

Nonetheless, five AR episodes occurred and were classified as grade 2R by the pathological characterization. Concurrent with the AR episodes, TAC concentrations in the EMBs were low at the three-month post-transplant time point (when three of the five episodes occurred); indeed, the corresponding median TAC_EMB_ concentration reached the lowest value (Figure 5). Patients’ blood-type mismatches and total ischemic times cannot be considered confounding factors, as reported in Table 2; transplantation with blood-type mismatch is never performed even in emergency cases and the total ischemic time was shorter than 4 h for all three patients. Figure 5 also confirms that the median concentration–time profiles of TAC in EMBs in the rejecting and non-rejecting patient groups are different with the median concentrations being lower for patients suffering from rejection. Completely different and unexpected results were obtained by analyzing data from whole blood (Figure 6).

Two patients died 1 and 3 months after transplantation, respectively. The cause of death was related to HTx in the first patient who died from multiple organ failure (MOF). The cause of the second death was instead intracerebral hemorrhage that occurred in a patient without hypertension.

As the transplanted patients and the corresponding donors are different individuals with different genetics, investigations of the role of the polyglycol protein (P-gp) directly in the graft will be carried out at this center next year. P-glycoprotein is a transmembrane glycoprotein that is directly encoded by the human ABCB1 gene. It is responsible for the efflux of many harmful compounds inside the cell to the extracellular space, but on the other hand, it also favors the removal of many drugs from the cells leading to a substantial reduction in their activity. P-glycoprotein controls drug absorption, distribution, and elimination in the body. Previous recent studies report that both TAC and CyA are substrates of P-gp; this has been demonstrated mainly for liver and kidney transplant recipients [19,20]. In 2002, Messner and colleagues described the expression and localization of the P-gp in fifteen left ventricular samples and observed a very wide inter-individual variability [21]. Future areas of investigation will address the characteristics of the donors in terms of the expression and localization of P-gp in the myocardial tissue to confirm or improve on Messner’s results.

We have not reported our results on tacrolimus concentrations in peripheral blood mononuclear cells because the trend is often in good agreement with its corresponding concentrations in whole blood.

In our opinion, all our future data combined with standardized pharmacokinetic analysis will be a requirement to achieve personalized therapy. Some individual characteristics could be strongly related to the mechanism of action of the drug and may reflect the personal response to the treatment; the synthesis of clinical signs and biological and histological parameters would allow both the minimization of immunosuppressive therapy and an improvement in the outcomes.

To the best of our knowledge, this is the first work reporting results of TAC concentrations in EMBs from HTx patients.

This study argues that TAC tissue concentrations in the allograft cannot be accurately predicted based on the blood level and that this is a possible mechanism underlying the AR occurrence. However, further studies and a larger population are needed to confirm these findings. In consideration of the need for cardiac transplant recipients to be closely monitored with clinical and imaging methods to early diagnose AR despite whole blood immunosuppressant concentrations within the therapeutic range, the routine implementation of analytical procedures to identify low allograft tissue levels will allow for more personalized therapeutic regimens, a step forward to AR defeat and a reduction of drug toxicities.

## Figures and Tables

**Figure 1 pharmaceutics-14-01247-f001:**
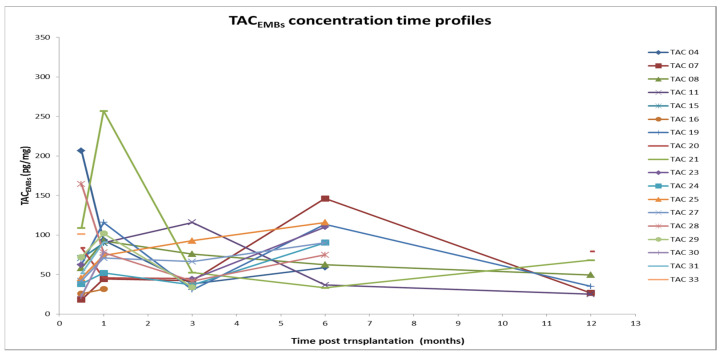
Concentration–time profiles of TAC in EMBs (18 patients). TAC: Tacrolimus; EMB: Endomyocardial biopsy.

**Figure 2 pharmaceutics-14-01247-f002:**
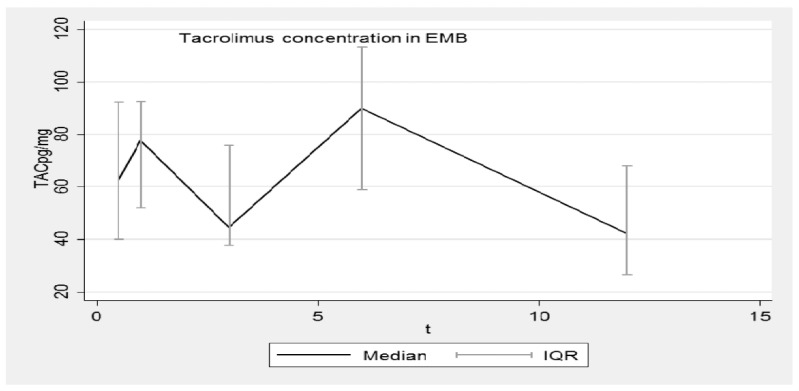
Median concentration–time profile of TAC in EMBs. TAC: Tacrolimus; EMB: Endomyocardial biopsy; IQR: interquartile range.

**Figure 3 pharmaceutics-14-01247-f003:**
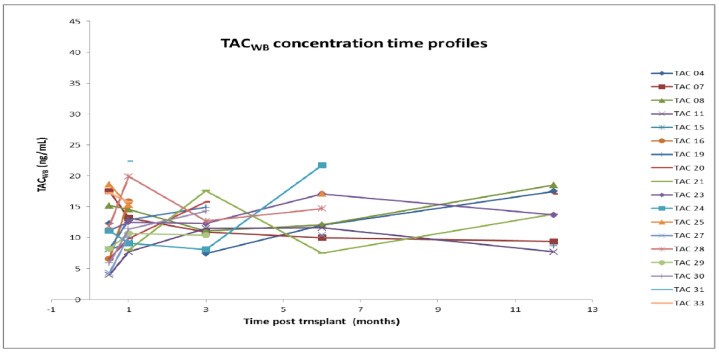
Concentration–time profiles of TAC in whole blood (18 patients). TAC: Tacrolimus; WB: whole blood.

**Figure 4 pharmaceutics-14-01247-f004:**
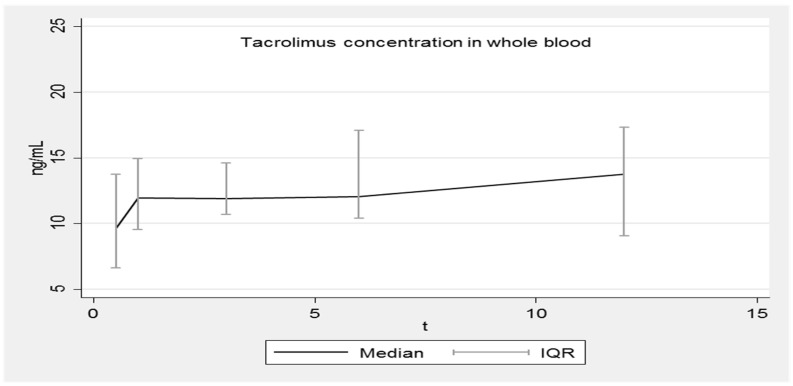
Median concentration–time profile of TAC in whole blood. IQR: interquartile range.

**Figure 5 pharmaceutics-14-01247-f005:**
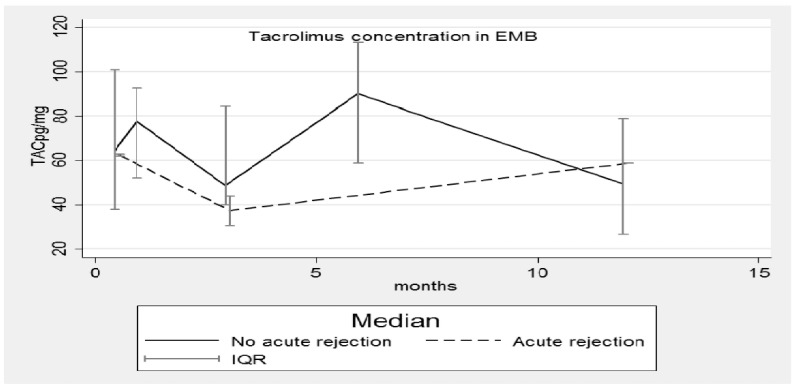
Median concentration–time profile of TAC in EMBs with and without acute rejection episodes. TAC: Tacrolimus; EMB: Endomyocardial biopsy; IQR: interquartile range.

**Figure 6 pharmaceutics-14-01247-f006:**
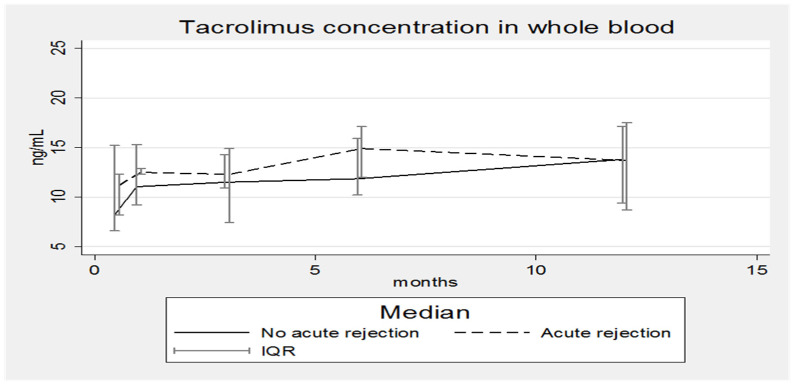
Median concentration–time profile of Tacrolimus in whole blood with and without acute rejection episodes. IQR: interquartile range. No correlation was found between TAC concentrations in EMBs and whole blood, considering all patients together as a single group and as subgroups of RPs or NRPs.

**Table 1 pharmaceutics-14-01247-t001:** TAC median concentrations in EMBs during the first year post-HTx.

Time Post-HTx	N° Analyzed EMBs	TAC Conc. pg/mg Median (Min–Max)
15 days	17	62 (18–207)
1 month	17	86 (31–257)
3 months	16	45 (31–119)
6 months	11	90 (33–146)
12 months	9	66 (25–112.5)

TAC: Tacrolimus; EMB: Endomyocardial biopsy; HTx: Heart transplant.

**Table 2 pharmaceutics-14-01247-t002:** Patients’ characteristics.

Patient ID	M/F	Age (Year)	Blood Type		Total Ischemic Time	Cold Ischemia Time	Warm Ischemia Time
			Donor	Recip			
TAC-23	M	57	A-	A-	140 min	77 min	63 min
TAC-19	F	54	A-	A-	95 min	50 min	45 min
TAC-04	F	58	AB+	AB+	188 min	130 min	58 min

**Table 3 pharmaceutics-14-01247-t003:** Tacrolimus concentration–time profiles in whole blood and EMBs.

	**Median**				
Months after HTx	0.5	1	3	6	12
N° analyzed EMBs	17	17	16	11	9
Whole blood (ng/mL)	8.2	12.0	12.5	13.4	13.8
EMB (pg/mg)	62.0	85.9	45.0	90.0	66.0
Ratio WB/EMB	0.13	0.14	**0.28**	0.15	0.21

EMB: Endomyocardial biopsy; HTx: Heart transplant; WB: whole blood.

## Data Availability

Not applicable.

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
