# Peer review of "Therapeutic Drug Monitoring of Tacrolimus-Personalized Therapy in Heart Transplantation: New Strategies and Preliminary Results in Endomyocardial Biopsies"

_pharmaceutics, 2022, doi:10.3390/pharmaceutics14061247_

Round 1

Reviewer 1 Report

In this manuscript, several concerns need to be addressed as follows:

  1. Lines 79: 81: transfer to the method section.
  2. Line 88-89: remove the title of the protocol.
  3. In the method section: clarify when the study has been performed (i.e. year …to year….).
  4. The statistical analysis: the authors should analyze the data with a suitable statistical model and not only represent the data as means and error. The authors should include the time post-transplantation as a factor to clarify whether the changes were significant or not.
  5. Results:
  • Lines 221-227: delete it as it is a repetition of the methods used.
  • The results text needs to be rewritten in a way reflecting the new statistical analysis and describing the findings as significant or non-significant.
  • The presentation of figures needs to be improved.
  • The full term of all abbreviations within figures and tables should be clarified.
  1. The discussion needs to be extensively revised as, in the present form, it is just a repetition of the introduction and methods.
  2. It is not preferred to begin sentences with abbreviations like TAC in lines 53 and 60. Please revise the whole manuscript for such an error.
  3. The manuscript needs to be revised for the English and the overall writing style. The writing style should be formal from the third-person perspective. Do not use we or our.
  4. There is a problem in using abbreviations throughout the manuscript. The full term should be mentioned first with the abbreviation between paresis then the abbreviations should be exclusively used throughout the manuscript. E.g., in line 40, Tacrolimus has been abbreviated as TAC then the full term has been repeated again in lines 155, 156, 222, 265, 340, and 347. Such errors have been repeated for many abbreviations throughout the manuscript.
  5. The conclusion is missed.

Author Response

Dear Reviewer 1,

Thank you for the helpful comments that provided us with guidance in improving our paper. As requested, we have modified the text following all your suggestions .

Hoping to have replied satisfactory to all concerns, we remain at your disposal for any other questions.

  • Lines 79: 81: transfer to the method section.

We transferred the lines 79-83 to the method section

  • Line 88-89: remove the title of the protocol

We removed the title of the protocol

  • In the method section: clarify when the study has been performed (i.e. year …to year….).

We better specified that the study started in 2020 and is still ongoing

  • The statistical analysis: the authors should analyze the data with a suitable statistical model and not only represent the data as means and error. The authors should include the time post-transplantation as a factor to clarify whether the changes were significant or not.

A joint modeling would be more appropriate considering the association between time-to-event (acute rejection episode) and the measured longitudinal data (TAC concentrations).

However, in our opinion, the limited sample size (5 events, 3 patients) is not adequate for estimating the effect of the longitudinal process in a joint modeling (Liddy M. Chen,Joseph G. Ibrahim, Haitao Chu. Sample size and power determination in joint modeling of longitudinal and survival data. https://doi.org/10.1002/sim.4263)

We reported this consideration in the 2.7. Statistical analysis paragraph

  • Lines 221-227: delete it as it is a repetition of the methods used.

We deleted the suggested lines

The results text needs to be rewritten in a way reflecting the new statistical analysis and describing the findings as significant or non-significant.

Please see point 4

The presentation of figures needs to be improved.

We changed and improved the “Results” paragraph.

The full term of all abbreviations within figures and tables should be clarified.

We clarified the full term of all abbreviations within figures and tables

  • The discussion needs to be extensively revised as, in the present form, it is just a repetition of the introduction and methods

We extensively revised the discussion.

  • It is not preferred to begin sentences with abbreviations like TAC in lines 53 and 60. Please revise the whole manuscript for such an error.

We revised the whole manuscript for the suggested error.

  • The manuscript needs to be revised for the English and the overall writing style. The writing style should be formal from the third-person perspective. Do not use we or our.

The whole manuscript was reviewed by a certified agency.

  • There is a problem in using abbreviations throughout the manuscript. The full term should be mentioned first with the abbreviation between paresis then the abbreviations should be exclusively used throughout the manuscript. E.g., in line 40, Tacrolimus has been abbreviated as TAC then the full term has been repeated again in lines 155, 156, 222, 265, 340, and 347. Such errors have been repeated for many abbreviations throughout the manuscript

We checked the whole manuscript following the suggestions.

  • The conclusion is missed.

According to the journal’s  Instructions for Authors, (Conclusions: This section is not mandatory but can be added to the manuscript if the discussion is unusually long or complex) we chose to report our conclusions and discussion in the same paragraph “Discussion”.

Reviewer 2 Report

Title: “Therapeutic Drug Monitoring of Tacrolimus-Personalized Therapy in Heart Transplantation: new strategies and preliminary results in endomyocardial biopsies.”

Authors: Simona De Gregori, Annalisa De Silvestri, Barbara Cattadori, Andrea Rapagnani, Riccardo Albertini, Elisa 5 Novello, Monica Concardi, Eloisa Arbustini and Carlo Pellegrini

Affiliation: Centro Malattie Genetiche Cardiovascolari- Fondazione IRCCS Policlinico San Matteo, Milan, Italy

Premise: Protocol routine scheduled (not for cause) endomyocardial biopsies (EMBs) may offer better monitoring than trough whole blood tacrolimus (TAC) monitoring for cardiac allograft monitoring.

Significance: To the best of their knowledge, this is the first work showing preliminary results of TAC concentration profiles (pg/mg) in EMBs of HTx patients, at six scheduled follow-up visits during the first post-transplant year.

Standard immunosuppression of Steroid, TAC and MMF is utilized in their program. To date 33 patient have been enrolled and 18 have undergone HTx and 15 are on a waiting list. Routine EMBs and whole blood TAC levels  (n=70) were performed at 0.5, 1, 3, 6, and 12 months. Whole blood [TAC] was measured by antibody-conjugated magnetic immunoassays (ACMIAs) on Siemens Dimension instrument. EMBs cardiac tissue [TAC] was measured by mass-spectrometry after enzymatic digest of know Bx mass and volume of standard digestion buffer. Known amount of FK506 with two deuterium atoms located on the carbon-13 was added to each digest vial sample for internal standard to quantify tissue [TAC]. Standard curves were generated with swine heart tissue and know [TAC].

Major Concerns: This is a small sample size to report on a basically negative study. We do not know the size of the Heart Transplant program and how many heart transplants are performed per year which has been shown to effect outcomes. We do not know cold or warm ischemic times, nor do we know if any HTx were blood group ABO incompatible that could have confound this small sample size. There is no discussion on the bioavailability of TAC. It is lipophilic, poorly absorbed from the intestine, widely distributed with a volume of 1,342 L and a significant first pass clearance. With its’ erratic absorption from the gut most centers aim for higher doses in the first 6 months. The authors could have referred to the preclinical studies conducted by Thomas Starzl and Fujisawa demonstrating tissue uptake for reference levels. The discussions is very limited. It is basically a repeat of the data already presented.

Minor Concerns: Many transplant centers target TACwb level greater than 10ng/ml for the first six months. We don’t know if this is BID or once daily dosing of TAC which increases compliance. Why not use the same mass-spectrometry for measuring [TACwb]?

Line by line critique:

Page 2, line 45; “…100 times lower than Cya and…” typo ‘CyA’

Page 4, line 177; “…used a rigid biotome…” typo suggest ‘used a rigid bioptome…’

Page 4, line 189;  “After extraction of the tissue fragment the cardiologist was careful not to remove the specimen with forceps…..” Note on the same page, line 176, the authors state that “at their center, the same cardiothoracic surgeon, performed the transvenous biopsies.” So, what is it Cardiologist or CV Surgeon? Or is it a team, and the cardiologist is helping by removing the tissue sample from the bioptome?

Page 4, line 192; “…fixative should be at room temperature to prevent additional contraction…” syntax/typo error suggest ‘…fixative was kept at room temperature to prevent additional contraction…’ also, wouldn’t needling the specimen induce a contraction?

Page 5, line 196; “…determination of the Tac concentration…” why change up the abbreviation for Tacrolimus? Suggest keep it constant at TAC or Tac.

Page 6, line 239, Table 2;  The Y-axis (vertical) is miss labeled for TAC whole blood (wb) levels (pg/mg) is for tissue as table 1 and for TACwb should be ng/ml.

Page 7, line 250; “…IQR…”Suggest spell out abbreviations in first instance ‘…interquartile range (IQR)…’

Page 8, line 278, Figure 5; Y-axis mislabeled with spacing issues and lack of units.

Page 10, line 320; “…the other from not defined intracerebral haemorrhage (ICH).”  Language suggest “…the other from undefined intracerebral haemorrhage.’ Probably don’t need abbreviation if not repeated.

Author Response

Dear Reviewer 2,

Thank you for the helpful comments that provided us with guidance in improving our paper. As requested, we have modified the text following all your suggestions.

Hoping to have replied satisfactory to all concerns, we remain at your disposal for any other questions.

Major Concerns:

This is a small sample size to report on a basically negative study. We do not know the size of the Heart Transplant program and how many heart transplants are performed per year which has been shown to effect outcomes. We do not know cold or warm ischemic times, nor do we know if any HTx were blood group ABO incompatible that could have confound this small sample size. There is no discussion on the bioavailability of TAC. It is lipophilic, poorly absorbed from the intestine, widely distributed with a volume of 1,342 L and a significant first pass clearance. With its’ erratic absorption from the gut most centers aim for higher doses in the first 6 months. The authors could have referred to the preclinical studies conducted by Thomas Starzl and Fujisawa demonstrating tissue uptake for reference levels. The discussions is very limited. It is basically a repeat of the data already presented.

Dear reviewer, the sample size is small because we reported the preliminary results (18 patients) of a whole program that requires 25 de novo transplant recipients (line 89). The number of heart transplants per year is not easily predictable, but we suppose the study will be completed within 3 years.

Warm or cold Ischemic (????), ABO blood group

Dear reviewer we added Table 2 to the manuscript, summarizing all your requested information. We explained the results within the document.

We improved the text with some tacrolimus characteristics, but, the choice for TAC dose administration after HTx are reported in line 309-311.

The discussions is very limited. It is basically a repeat of the data already presented.

We changed and improved the discussion paragraph: wee better declared our results, limitations (limited number of EMBs) and proposed new approaches to deeper investigate factors that could contribute to acute rejection episodes. In the next years, when the study will be finished, I’m sure we will provide deeper observations and conclusions.

Minor Concerns

Many transplant centers target TACwb level greater than 10 ng/ml for the first six months.

The therapeutic range of tacrolimus has not been clearly defined, but some researchers report a range of 5 to 20 µg/L in whole blood, [1] while others suggest that 5 to 15 µg/L may be more appropriate [2]. Generally, high tacrolimus concentrations are likely to be required in the initial post-transplant period, but target concentrations can then be reduced over time.

On the contrary, as reported in the “Discussion” paragraph, the physicians chose to administer a low dose of Tacrolimus in the first days after HTx in order not to overload the kidneys of patients already being treated with nephrotoxic drugs.

1] Jusko WJ, Thomson AW, Fung JJ, et al. Consensus document: therapeutic monitoring of tacrolimus (FK506). Ther Drug Monit 1995; 17: 606-14

2] McMaster P, Mirza DF, Ismail T, et al. Therapeutic drug monitoring of tacrolimus in clinical transplantation. Ther Drug Monit 1995; 17: 602-5

We don’t know if this is BID or once daily dosing of TAC which increases compliance.

Patients are being treated with Tacrolimus twice daily dosing (BID). We added this information in “Materials and Methods – Patients section.

Why not use the same mass-spectrometry for measuring [TACwb]?

We measured TACwb both by antibody-conjugated magnetic immunoassay (ACMIA) and HPLC-MS/MS (data not reported): we noticed, as expected, a good correlation between the two analytical methods, but we only reported ACMIA results that are routinely communicated to Physicians as a therapeutic TAC monitoring.

Line by line critique:

Page 2, line 45; “…100 times lower than Cya and…” typo ‘CyA’

We replaced Cya with CyA

Page 4, line 177; “…used a rigid biotome…” typo suggest ‘used a rigid bioptome…’

We replaced biotome with bioptome

Page 4, line 189;  “After extraction of the tissue fragment the cardiologist was careful not to remove the specimen with forceps…..” Note on the same page, line 176, the authors state that “at their center, the same cardiothoracic surgeon, performed the transvenous biopsies.” So, what is it Cardiologist or CV Surgeon? Or is it a team, and the cardiologist is helping by removing the tissue sample from the bioptome?

The same cardiovascular surgeon performs the transvenuos biopsies and removes the tissue sample from the bioptome.

We corrected the text.

Page 4, line 192; “…fixative should be at room temperature to prevent additional contraction…” syntax/typo error suggest ‘…fixative was kept at room temperature to prevent additional contraction…’ also, wouldn’t needling the specimen induce a contraction?

We changed the text, following your suggestions.

The procurement and processing of biopsy samples may result in histologic artifacts which should not be interpreted as pathologic changes. Contraction bands are frequently an artifact of the biopsy procedure itself and do not necessarily indicate myocardial ischemia. In contrast to necrosis with contraction bands, artefactual contraction bands often extend across many myocytes which are otherwise normal. Fixing the specimens at room temperature, rather than in chilled fixative, will reduce this artifact.

Page 5, line 196; “…determination of the Tac concentration…” why change up the abbreviation for Tacrolimus? Suggest keep it constant at TAC or Tac.

We replaced Tac with TAC

Page 6, line 239, Table 2;  The Y-axis (vertical) is miss labeled for TAC whole blood (wb) levels (pg/mg) is for tissue as table 1 and for TACwb should be ng/ml.

We replaced (pg/mg) with ng/mL

Page 7, line 250; “…IQR…”Suggest spell out abbreviations in first instance ‘…interquartile range (IQR)…’

We spelled out ‘……interquartile range (IQR) in line 242.

Page 8, line 278, Figure 5; Y-axis mislabeled with spacing issues and lack of units.

We replaced Fig.4 and Fig. 5

Page 10, line 320; “…the other from not defined intracerebral haemorrhage (ICH).”  Language suggest “…the other from undefined intracerebral haemorrhage.’ Probably don’t need abbreviation if not repeated.

We changed the text following your suggestions.

Round 2

Reviewer 1 Report

No further comments to be addressed